

# Understanding the impact of data gaps on long-term offshore wind resource estimates

Martin Jonietz Alvarez[a], Warren Watson[a], and Julia Gottschall[a]

[a]Fraunhofer Institute for Wind Energy Systems IWES, Am Seedeich 45, 27572 Bremerhaven, Germany

**Correspondence:** martin.georg.jonietz.alvarez@iwes.fraunhofer.de

**Abstract.** In the context of a wind farm project, the wind resource is assessed to predict the power output and the optimal positioning of the wind turbines. That requires taking wind measurements on the site of interest and extrapolating these to the long-term using so-called "measure, correlate, and predict" (MCP) methods. The failure of sensors, power supply, or software are common phenomena. These disruptions cause gaps in the measured data, which can be especially long in offshore
measurement campaigns due to harsh weather conditions causing system failures and preventing servicing and redeployment. The present study investigates the effect of measurement data gaps on long-term offshore wind estimates by analyzing the bias they introduce in the parameters commonly used for wind resource assessment. Furthermore, it aims to show how filling the gaps can mitigate their effect. To achieve this, we perform the investigations for three offshore sites in Europe with 2 years of concurrent measurements. We use reanalysis data and various MCP methods to fill gaps in the measured data and extrapolate
this data to the long term. The results of the investigations show that the effects of gaps on long-term extrapolations are lower than expected. For instance, gaps of 180 days cause an average deviation of the long-term mean wind speed of less than $0.04\,\mathrm{ms^{-1}}$ for all tested sites. Filling the gaps can slightly reduce their impact if the MCP method used for gap filling performs better for predicting known data than the MCP method used for long-term extrapolating.

## 1 Introduction

Reliably predicting the wind speed and wind direction is necessary to analyze a potential wind farm site and to lower the economic investment risks associated with the project. These forecasts are based on data that is collected on the pre-selected site. The wind resource assessment accuracy increases with the amount of data on-site data available. Nevertheless, the cost of the measurement campaign increases with its duration. Therefore, wind farm operators resort to measurement campaigns of 1 to 2 years to save time and costs. These measurements are then extrapolated to the whole expected lifetime of the wind farm,
which usually reaches between 20 to 30 years (Rohrig et al., 2017).

The long-term extrapolation (LTE) is done by determining a correlation function that describes the relationship between the measurement and an available reference data set that gives a long-term record of the meteorological conditions from a nearby site. This correlation is established over the training period, in which both data sets are available. In the second step, the correlation function is applied to the reference data in the target period (period in which there is no measurement available).





The methods that follow this principle are known as "measure, correlate, and predict methods" (MCP methods) (MEASNET, 2016).

For industry applications, the most commonly used MCP method is based on linear regression through concurrent measurement and reference data points (Carta et al., 2013). This procedure can be extended by doing a different regression for each wind direction sector or each wind speed range. Furthermore, MCP methods using multiple regression functions (Beltran et al.,
2010), probabilistic distributions (Borujeni et al., 2021), and multiple reference data sources (Carta et al., 2013) have been proposed. Hanslian (2017) classifies the MCP methods into two types: type 1 methods, which are based on time series corrections such as linear regressions and excel for estimating time series, and type 2 methods, which take a probabilistic approach so that they suit the prediction of wind speed distributions and average values. The works of Schwegmann et al. (2023) and Borujeni et al. (2021) show that various machine learning algorithms can be used as MCP methods as well. Among the algorithms tested
by Schwegmann et al. (2023), the K-Nearest-Neighbors (KNN) regression method performs best for doing one-day wind speed predictions and is recommended for further applications.

A significant proportion of the wind data collected in measurement campaigns is erroneous or unavailable due to measurement equipment failures or external disturbances. These issues generate gaps that can extend up to various months in the data sets that are to be extrapolated to the long term. This increases the uncertainty associated with long-term predictions. Therefore,
the current industry requirements on the data availability of a measurement campaign for wind resource assessment surpass the 80% (FGW, 2020) or even the 90% mark (MEASNET, 2016). Other purposes such as verification of floating lidar measurement systems require availabilites surpassing 95% (OWA, 2018). Common solutions for preventing or compensating for data losses are robust measurement set-ups, device monitoring, redundancies, and measurement campaign extensions. These measures are either costly or decrease the suitability of the data set for wind resource assessment. For instance, extending the
campaign leads to inhomogeneous data coverage across the seasons, leading to a biased wind resource assessment.

Gaps can be filled to increase the measurement availability and reduce errors on wind resource assessments. A simple approach for filling single-value gaps by interpolating through adjacent time stamps is studied by Pappas et al. (2014). Another option for filling gaps in wind speed and direction measurements is to extrapolate data from other heights by physical wind profile modelling (Landberg, 2015) or by using machine-learning based methods (Rouholahnejad et al., 2023). If the gaps are
longer than a single value and no measurements from other heights are available, using an MCP method is an option to fill the data gaps as well (MEASNET, 2016). Nevertheless, Gottschall and Dörenkämper (2021) show that filling the gaps does not always mitigate their effect on an LTE, as this effect is already very small, with a 30-day gap causing approximately a $0.01\,\mathrm{ms}^{-1}$ error on average on the long-term mean wind speed.

The present study builds on the results of Gottschall and Dörenkämper (2021) about the effect of gaps on long-term extrapo-
lations, extending the investigated cases to measured data with longer and multiple gaps. Furthermore, the impact of the choice of the gap-filling MCP method on the LTE is investigated by analyzing the mitigation of the gap effect for different gap-filling methods (including the method used by Gottschall and Dörenkämper (2021)). For the present investigations, we use the same data as Gottschall and Dörenkämper (2021) to ensure the comparability of results. The focus of the present study lies only on wind speed and wind direction data, although results may be valid for other atmospheric parameters as well.





The present work is structured in 6 sections including this introduction. Section 2 describes the met mast data from different sites taken as measurement data and the ERA5 reanalysis data for the locations of the met masts taken as reference data for the MCP methods. Section 3 includes a description of the artificial gap introduction into the measurement data, an overview of the MCP methods used, and a description of the method used to measure the effects of gaps and gap filling on long-term extrapolations. Section 4 presents the results of the main addressed research questions:

- How do the three implemented MCP methods (Linear Regression, Sector Average Deviation addition, and KNN regression) compare to each other when filling artificial data gaps?

- Is there a correlation between the length of a measurement data gap and its effect on the long-term extrapolation? How does this correlation change for multiple gaps instead of one?

- Under which circumstances does gap filling mitigate the gap effect on long-term extrapolations?

The results of the research questions are discussed in Section 5. Finally, in Section 6, the conclusions of the present study are summarised.

## 2 Data basis

Offshore met mast measurements at three sites and numerical data obtained for the closest available location to each of the met masts are the data sources used for the investigations of the present study. These sites represent various offshore conditions in the North and Baltic Sea. We selected the same data sets as Gottschall and Dörenkämper (2021), as we build the present investigations on their studies.

### 2.1 Sites and met mast measurement data

In the present study, only met mast measurements are used as training data for the MCP methods, as it captures the exact on-site conditions. The data sets from three met masts Ijmuiden, FINO2 and FINO3 are pre-processed and used. Therefore, various surrounding conditions are considered as the sites have different distances to the coast and atmospheric stabilities. These three measurement data sets are publicly available for research purposes. Their positions can be seen in Figure 1.

The measurements from the met masts are all taken with cup or sonic anemometers and wind vanes. The 10 minute averaged values of horizontal wind speed and wind direction are used. The values from the respectively closest sensors to 90 m above the mean sea level are taken since this is a common hub height of offshore wind turbines. A simultaneous measurement period of 24 months is considered in the present work: from 01-07-2012 to 30-06-2014. We select this period because of the high data availability of the three sites and because no wind farms were erected or decommissioned nearby during that time (Gottschall and Dörenkämper, 2021).

The data sets of the three masts contain gaps. In the pre-processing step, these gaps are filled before the application of the methodology described in Section 3. To complete the wind direction time series, measurements from sensors of lower heights



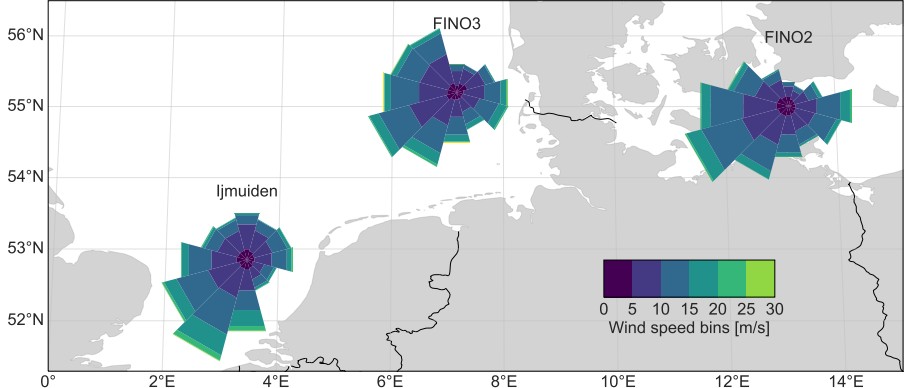

**Figure 1.** Position of the sites FINO2, FINO3 and Ijmuiden including wind roses for the pre-processed met mast data for the period from 01-07-2012 to 30-06-2014. Adapted from Meyer and Gottschall (2022).

are used. The missing wind speed values are taken from lower heights as well, in this case multiplying with a factor to account for the wind profile as described by Gottschall and Dörenkämper (2021). To reach a 100% measurement data availability in all sites, the remaining gaps are filled using the KNN MCP method. For this, we set the K-parameter to 200 points for wind speed and 700 points for wind direction. A detailed description of the used KNN method can be found in Section 3.1.3. The KNN-filling is the only difference between the input data used in the present study and the data used by Gottschall and Dörenkämper

(2021). Following specifications apply to the met masts and the sites:

- The **Ijmuiden** met mast is located in the North Sea at approximately 75 km West of the coast of Ijmuiden (coordinates: 52°51'00"N 3°26'24"E). It provides measurements at several heights and is described in more detail by Poveda et al. (2015). For the analysis in the present study, the wind speed measurement at 92 m (cup anemometer) and wind direction measurement at 87 m are used. For the measurement period and heights used, the mean wind speed at this site is 9.88 ms$^{-1}$

and the mean wind direction is 233.4°.

- The **FINO2** met mast is situated in the central southern Baltic Sea (coordinates: 55°00'25.2"N 13°09'14.4"E), thus being affected by distances to land of less than 50 km in most directions. FINO2 provides wind measurements at various heights between 32 m and 102 m above sea level, technically described in FINO2 (2007). The wind speed and direction measurements from the sensors mounted at 92 m altitude (cup anemometer and vane, respectively) are considered in the

present investigations. For the measurement period and heights used, the mean wind speed at this site is 9.59 ms$^{-1}$ and the mean wind direction is 228.3°.

- **FINO3** is a met mast located in the North Sea, 80 km West of Sylt (coordinates: 55°12'00.0"N 7°09'36.0"E). No land influences the main wind direction sectors from South to Northwest. FINO3 (2012) can be referred for further technical Information about this met mast. Leiding et al. (2012) offers detailed descriptions and data analyses of the FINO mea-

surements. The measurements of wind speed and direction from the sensors at the heights of 92 m and 101 m respectively





are used in the present work. For the measurement period and heights used, the mean wind speed at this site is $9.60 \, \text{ms}^{-1}$ and the mean wind direction is 243.6°.

As the reanalysis data used as reference for the MCP methods is only available with hourly resolution, we only use measurements recorded at whole hour time stamps.

## 2.2 Reanalysis reference data

In contrast to the generally short-term and expensive met mast measurements, reanalysis data is available globally, for periods reaching back to the year 1950, and without gaps. Nevertheless, reanalyses are currently not capable of capturing weather conditions at a specific location as accurately as a met mast or a lidar due to their limited spatial and temporal resolution. Therefore, the sole use of reanalysis data for wind site assessment is disregarded. Nevertheless, it is still of common use in the industry as reference data for MCP methods (Gottschall and Dörenkämper, 2021). The fifth major global reanalysis (ERA5) is the most recent generation of reanalysis data issued by the European Center for Medium-range Weather Forecasts (ECMWF) (Hersbach et al., 2020). The time resolution for the atmospheric parameters is one hour, and the spatial resolution is 0.25° in latitude and 0.25° in longitude. For each met mast, we take the ERA5 data set that results from the spatial bi-linear interpolation between the four grid points that are closest to the met mast location. The period between the years 1994 and 2014 is used, as this is the target period of the long-term extrapolations done in the present work.

We are aware of the existence of other sources with higher spatial and temporal resolutions that can be taken as references for the MCP methods as well. Some examples of this are the data from the New European Wind Atlas (NEWA) (Dörenkämper et al., 2020) and the mesoscale modeling data optimized with the Weather Research and Forecasting (WRF) (Gottschall and Dörenkämper, 2021). Nevertheless, both of these sources have a worse correlation to the met mast measurements used in the present work than ERA5 data, which makes them less suitable as references for the MCP methods (Meyer and Gottschall, 2022). Therefore, in the present study, ERA5 is used as reference data for the MCP methods for gap-filling and long-term extrapolating.

The ERA5 reference data sets show a strong linear correlation with the respective met mast measurements. The coefficient of determination ($R^2$) between ERA5 and met mast wind speed data is 0.89 for the sites of FINO2 and FINO3 and 0.93 for the site of Ijmuiden.

## 3 Applied methodology

The methods used to introduce artificial gaps into the met mast measurements, fill these gaps, and extrapolate the original, gapped, and filled data sets to the long term are described in the following subsections. Furthermore, the methods used to evaluate the performance of MCP methods and the effect of gaps on long-term extrapolations are presented.





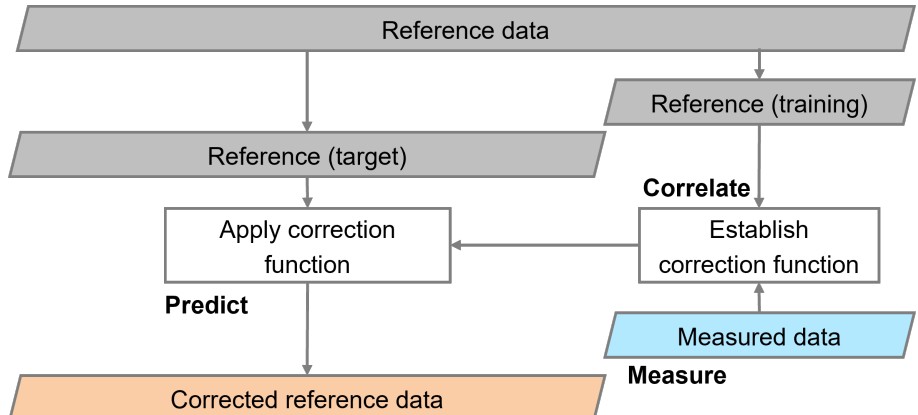

**Figure 2.** Workflow of a time series correction MCP method used to obtain a corrected reference time series. Reference data is marked in gray, measured data in blue and corrected reference data in orange.

## 3.1 MCP methods

In the present study, met mast measured data sets are extended in time, either to fill data gaps or to extrapolate them to the long term. For these applications, we use the following MCP methods:

- Sector-wise Linear Interpolation (SLI) as described in Section 3.1.1.

- Sector Average Deviation (SAD) as described in Section 3.1.2.

- K-Nearest-Neighbors regression (KNN) as described in Section 3.1.3.

The three implemented MCP methods correct the reference time series when applied. MCP methods that follow this principle are classified as type 1 methods by Hanslian (2017). According to (Hanslian, 2017), type 1 methods are most suitable for predicting time series while type 2 MCP methods are most suitable for predicting distributions. Type methods include the creation of a correlation function between the concurrent measured and reference data and the application of the correlation function to correct the reference time series of the target period. The KNN method has type 2 features, as it classifies the wind speed data before applying the correction. Figure 2 shows a flowchart of time series correcting MCP methods.

### 3.1.1 Sector-wise linear interpolation (SLI) MCP method

The most commonly used MCP method for extrapolating wind speed measurements to the long-term is the simple linear interpolation method (Carta et al., 2013). Given its widespread use and the high correlation between measurement and reference data in all investigated sites, we consider this method in the present work.





The correction is done separately for each 30° wind direction sector to account for inhomogeneous surrounding conditions. As recommended by Carta et al. (2013), we add a Gaussian noise term, although the option without a noise term is investigated in Section 4.1. In the following, this MCP method is abbreviated as SLI (Sector-wise Linear Interpolation).

### 3.1.2 Sector average deviation (SAD) MCP method

Generally, the long-term wind direction is assumed to be the same as the wind direction recorded over a year-long period at the same site (Carta et al., 2013). However, several studies reviewed by Carta et al. (2013) propose MCP methods for estimating the long-term wind direction. One of the most common wind direction MCP correction methods is the sector-wise deviation correction. It is used, for instance, in the studies of Gottschall and Dörenkämper (2021) and the SpeedSort method presented by King and Hurley (2005).

In the sector-wise deviation MCP method, the average wind direction deviation between the concurrent measured and reference data is calculated for each wind sector. These deviations are added to the reference data from the target period to obtain the MCP-corrected time series. In the present work, a sector size of 30° is chosen. We do not consider data pairs for which the reference wind speed value is less than $3\,\mathrm{ms^{-1}}$, as we estimate the wind direction measurements to be inaccurate fur such low wind speeds. A Gaussian noise term is applied to avoid empty wind direction ranges in between the sectors (the option without

a noise term is investigated in Section 4.1 as well). In the following, this MCP method is abbreviated as SAD (Sector Average Deviation).

### 3.1.3 K-Nearest-Neighbors (KNN) MCP method

For filling the gaps in Sections 4.1 and 4.3, we use the novel K-Nearest-Neighbors (KNN) regression MCP method additionally to the traditional MCP methods. In the studies of Schwegmann et al. (2023), the KNN method shows the best results among

several other MCP methods for filling gaps between 1 and 23 hours. Schwegmann et al. (2023) test the performance of several MCP methods by calculating the Root Mean Square Error (RMSE), the coefficient of determination ($R^2$), and the Jensen-Shannon distance between the original measurement and the MCP-filled data. We implement this algorithm using the Python package scikit-learn (Pedregosa et al., 2011). Contrary to the SLI and the SAD, we use this algorithm to correct wind speed and direction.

For correcting each reference data point from the target period, the KNN algorithm finds the K reference points from the training period with the least distance to the point to be corrected (K-nearest-neighbors). The distance between reference data points is calculated in the dimensions of the physical parameters included in the algorithm (also called features). A regression through the measurement values concurrent to the K-nearest-neighbors gives the target value (Cover and Hart, 1967; Schwegmann et al., 2023).

Schwegmann et al. (2021) found the feature combination that results in the lowest RMSE between the filled and the original wind speed time series: wind speed, wind direction, surface pressure, surface latent heat flux, sea surface temperature, and the temperature difference between the sea surface and 2 meters above sea level. These features are used as well in the present work. Furthermore, several settings influencing the KNN regression algorithm (also called hyperparameters) are specified.





**Table 1.** Mean and standard deviation (STD) of K for which the RMSE between the originally measured and the KNN-filled data is minimized. Values are obtained for 30-day gaps. Results for wind speed and wind direction for all investigated sites. Values are rounded to the closest integer

| Site | Wind speed mean | Wind direction mean | Wind speed STD | Wind direction STD |
| --- | --- | --- | --- | --- |
| FINO3 | 343 | 1950 | 374 | 3895 |
| FINO2 | 800 | 931 | 2455 | 2538 |
| Ijmuiden | 543 | 1661 | 923 | 3322 |

Schwegmann et al. (2021) select the hyperparameter combination that gives the lowest RMSE between original and predicted data for a testing subset. In the present work, hyperparameters are selected as follows:

– Number of neighbors (K): the K that results, on average, in the lowest RMSE between filled and original measurement data when filling single 30-day gaps with shifting gap starting date (see Subsection 3.3.1) is used.

– Dimension of the distance calculation in the feature space: the two-dimensional (Euclidean) distance is used.

– Weighting of the features: Uniform weighting is used.

– Other hyperparameters affecting computation time, such as leaf size: default values of scikit-learn are taken.

The 1-dimensional Nelder-Mead simplex algorithm (Arora, 2017) is used to find the optimum K for each site. Table 1 lists the average and the standard deviation across all introduced gaps of the optimum K for predicting wind speed and direction.

In Table 1, the standard deviation is higher than the average for every parameter and site. Therefore, the optimal selection of K varies drastically depending on the data that is predicted and no optimum exists for the general case. However, we use the mean values listed in Table 1 when filling the gaps with the KNN method, as they are estimates of the optimal K.

### 3.1.4 MCP method performance metrics

There are multiple ways to evaluate the performance of an MCP method. One approach is calculating the statistics that compare the original measured with predicted data value-by-value. The RMSE (Schwegmann et al., 2023; Hanslian, 2017) and the $R^2$ (MEASNET, 2016; Schwegmann et al., 2023) are the most used statistics for this. Nevertheless, long-term extrapolations aim to estimate overall statistics, such as long-term mean wind speed and long-term wind speed distribution. For this reason, comparing the statistics of the predicted and the original measured data is a method used in many studies as well (Gottschall and Dörenkämper, 2021; Hanslian, 2017; Schwegmann et al., 2023).

For the present studies, we use three overall time series statistics and two value-by-value statistics to evaluate the performance of the SLI, SAD, and KNN methods:

– Absolute mean wind speed (MWS) difference between predicted and original data (overall statistic).





– Wind speed distribution error (DE) between predicted and original quantified by the chi-squared test with wind speed bins of $1\,\mathrm{ms^{-1}}$ (overall statistic).

– Absolute mean wind direction (MWD) difference between predicted and original data (overall statistic).

– RMSE between predicted and original wind speed data (value-by-value statistic).

– RMSE between predicted and original wind direction data (value-by-value statistic).

The performance is considered better for all metrics the lower the value of the performance statistic is. In Section 4.1, we calculate these statistics for 105 artificially introduced 120-day gaps with different start dates (see Section 3.3.1 for more details on gap introduction). The averages of the MCP performance statistics over all introduced gaps are taken as the criteria to describe the performance of each MCP method.

## 3.2 Long-term extrapolations

In the present work, we calculate long-term extrapolations (LTE) to a target period of 20 years between 01-07-1994 and 30-06-2014, with ERA5 as reference data. The following Subsections contain the details on the MCP method used for long-term extrapolating and the description of how the gap effect on long-term extrapolations is evaluated.

### 3.2.1 Long-term extrapolation MCP method

MCP methods based on linear regression are the most commonly used for extrapolating measured wind speed time series to the long term. The sector average deviation MCP method (refer to Section 3.1.2) is commonly employed for wind direction extrapolation (Carta et al., 2013). Therefore, we use the SLI and SAD MCP methods for the long-term extrapolations to obtain the results shown in Sections 4.2 and 4.3. For the LTE, both MCP methods include the Gaussian noise term.

In the present work, the gaps are included in the target period when doing a long-term extrapolation of a gapped data set.
Therefore, the gaps are filled with the corrected long-term reference time series obtained when extrapolating. Figure 3 shows a schematic representation of this process.

The gaps are filled before the long-term extrapolations for calculating the results for the LTE of filled data shown in Section 4.3. In these cases, the filled gaps are considered part of the measurement data and belong to the training period instead of the target period.

### 3.2.2 Evaluation of the gap effect on long-term extrapolations

The energy yield of a wind turbine is calculated using the wind speed distribution. Therefore, this is the most relevant output of a long-term extrapolation, along with the mean wind speed (MWS) and mean wind direction (MWD) (MEASNET, 2016). Hence Gottschall and Dörenkämper (2021) consider that the gap effect on an LTE correlates with the mean wind speed, mean wind direction, and distribution deviations between the extrapolated gapped and original data. Therefore, we use the MWS
and MWD difference between the long-term extrapolated gapped and original data to quantify the effect of gaps on an LTE.





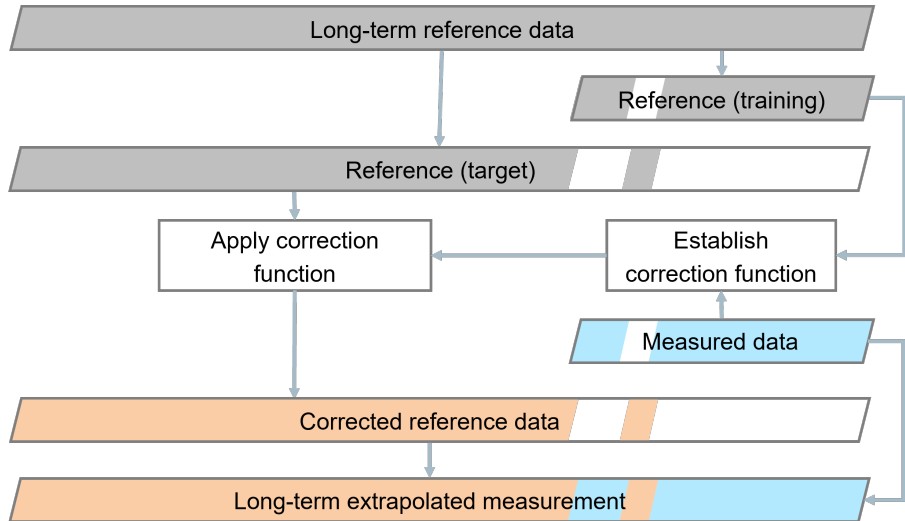

**Figure 3.** Workflow of a time series correction MCP method used for long-term extrapolation. Case with gapped measurement. Reference data is marked in gray, measured data in blue and corrected reference data in orange. Periods for which data is not available in the respective data set are marked in white. The long-term extrapolated measurement is composed of corrected reference and measured data.

Additionally, we use the wind speed distribution error (DE) between the long-term extrapolated gapped and original data to evaluate the gap effect on the LTE. We calculate the DE using the chi-square test with wind speed bins of $1\,\mathrm{ms^{-1}}$. Analogously, we quantify the effect of gap filling on an LTE through the deviations between the statistics of the extrapolated filled and original data.

For calculating the long-term MWS, MWD, and distribution, the extrapolated data set includes the measurement from the training period and the corrected reference data from the target period (see Figure 3). Analyses are repeated for gaps with different starting dates (see Section 3.3) for each gap duration investigated in Sections 4.2 and 4.3. To generalize the results over all gaps with the same gap duration introduced, we use the metric presented by Gottschall and Dörenkämper (2021): the RMSE between the gapped and the original long-term extrapolated MWS, MWD, and distributions. Similarly, we use the

RMSE between the filled and original long-term statistics to analyze the effect of gap filling on long-term extrapolations.

### 3.3   Gap generation

An artificially gapped data set is needed to evaluate the effect of gaps on a long-term extrapolation as described in Subsection 3.2.2. We do this by replacing the originally measured wind speed and direction values with non-numerical (NaN) values. A gap consists of one or multiple consecutive NaN values. We define each gap by setting the starting time stamp and the amount

the consecutive NaN values. The following subsections describe the two gap types investigated in the present work.





### 3.3.1 Single gap with shifting start date

The single gap introduction follows the procedure used by Gottschall and Dörenkämper (2021). It is composed of the following steps:

- Step 1: Introduce a gap with a defined starting date and duration into the measurement.

- Step 2: Derive a data set from the gapped data (filling the gap and/or extrapolating to the long term)

- Step 3: Calculate statistics of the derived data set (mean wind speed, distributions, etc...)

- Step 4: Shift the gap starting date forward by 7 days.

- Step 5: Repeat all previous steps with the shifted gap.

The gap is shifted through the 2-year measurement, resulting in a total of 105 gapped data sets analyzed. For gaps whose
ending date surpasses the end of the measurement, the corresponding number of time stamps is deleted at the beginning of the data set. In Section 4.1, results are shown for gaps with 120 days of duration. Sections 4.2 and 4.3 include gaps with durations ranging from 0 to 180 days in steps of 30 days. Considering a 2-year measurement, 180 days missing implies roughly a 75% data availability.

### 3.3.2 Multiple gaps with random start dates

We developed a process to introduce multiple gaps in a data set. These gaps represent unforeseeable availability losses in a more realistic manner than a single gap. This method includes the following steps:

- Step 1: Introduce multiple gaps with defined single and combined lengths into the measurement. The starting date of each gap is selected randomly.

- Step 2: Derive a data set from the gapped data (filling the gap and/or extrapolating to the long term)

- Step 3: Calculate statistics of the derived data set (mean wind speed, distributions, etc...)

- Step 4: Repeat all previous steps until the average over all repetitions of the statistics calculated in Step 3 converges.

For gaps that extend beyond the measurement ending timestamp, a corresponding number of NaN values is introduced at the start of the measurement time series. Overlapping gaps are merged into a single one with their combined length.

We use this method to analyze the effect of multiple gaps on long-term extrapolations in Section 4.2. For this, combinations
of gaps with total durations ranging from 0 to 180 days in steps of 30 days are introduced. Every single gap within those combinations is 30 days long, as this is a plausible time window between a failure of an offshore wind measurement system and its redeployment. Pre-defining the single and combined gap lengths constrains the number of introduced gaps.

For the investigations shown in Section 4.2, we require the convergence of the gap effect on long-term extrapolations. Therefore, the parameters presented in Section 3.2.2 must converge. We only use the long-term mean wind speed RMSE for





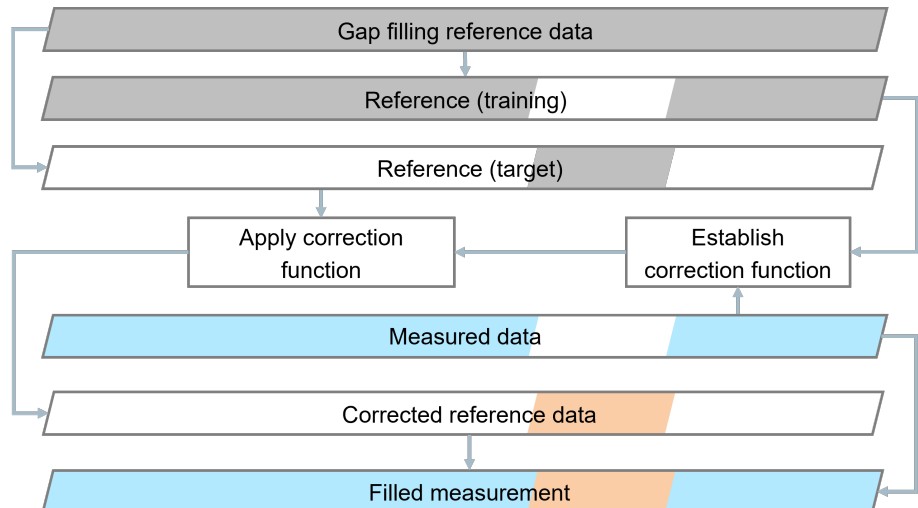

**Figure 4.** Workflow of a time series correction MCP method used to fill a measurement data gap. Reference data is marked in gray, measured data in blue and corrected reference data in orange. Periods for which data is not available in the respective data set are marked in white. The filled measurement is composed of corrected reference and measured data.

determining convergence. Accordingly, the repetition of Steps 1 to 4 is terminated when the long-term mean wind speed RMSE stays within a range of $0.0001\,\mathrm{ms^{-1}}$ for the last 100 gap combinations introduced. This criterion leads to stable results for all parameters describing the gap effect on long-term extrapolations.

### 3.4   Gap filling

The results shown in Sections 4.1 and 4.3 involve filling artificial gaps in the measured data using the MCP methods described

in Section 3.3. When filling a gap, the measurement-reference data pairs for the period outside of the gap are used for training and the reference data from the period covered by the gap is corrected. This procedure is illustrated in Figure 4.

### 4   Results

The following subsections show the results aimed at answering the research questions stated in the Introduction. We obtain these results using the data and the methods described in Sections 2 and 3.

**4.1   Comparison between MCP methods**

The average performance of each MCP method, analyzed as described in Section 3.1.4, is shown in Tables 2 to 4.

    On average, the KNN method performs slightly better than the SLI and the SAD methods (for wind speed and wind direction respectively) when measured by the RMSE over the gaps. By this metric, the performance of the MCP methods with a noise term is considerably worse than the performance of the same methods without a noise term. On the contrary, the distribution



**Table 2.** Performance statistics of the Sector-wise Linear Interpolation (SLI), Sector Average Deviation (SAD), and K-Nearest-Neighbors (KNN) MCP methods for the FINO3 site. The statistics shown are the Mean Wind Speed (MWS) and Mean Wind Direction (MWD) average deviations, the average Wind Speed Distribution Error (WS DE), and the average Wind Speed and Wind Direction Root Mean Squared Error (WS RMSE and WD RMSE). Deviations between the MCP-predicted and the original statistics are calculated over 120-day periods, with starting dates spaced by 7 days along the measured period. The lowest value of each column is written in bold characters.

| MCP method | MWS deviation [ms$^{-1}$] | WS DE [%] | MWD deviation [°] | WS RMSE [ms$^{-1}$] | WD RMSE [°] |
|---|---|---|---|---|---|
| SLI | 0.13 | 3.47 | - | 1.34 | - |
| SLI with noise term | 0.13 | 2.49 | - | 1.85 | - |
| KNN | **0.06** | **1.81** | **3.81** | **1.30** | **18.4** |
| SAD | - | - | 4.66 | - | 18.5 |
| SAD with noise term | - | - | 4.65 | - | 18.9 |

**Table 3.** Performance statistics of each MCP method for the FINO2 site. Labels of statistics and MCP methods as described in Table 2.

| MCP method | MWS deviation [ms$^{-1}$] | WS DE [%] | MWD deviation [°] | WS RMSE [ms$^{-1}$] | WD RMSE [°] |
|---|---|---|---|---|---|
| SLI | 0.17 | 3.06 | - | 1.44 | - |
| SLI with noise term | 0.16 | **2.60** | - | 2.01 | - |
| KNN | **0.12** | 2.71 | 1.82 | **1.42** | **19.0** |
| SAD | - | - | 1.55 | - | 19.1 |
| SAD with noise term | - | - | **1.54** | - | 19.5 |

error when using the SLI method is lower with than without noise term. It can also be noted that the MWS and MWD deviations are almost identical for the SLI and SAD options with and without a noise term. When comparing the performance of the KNN method to the performance of the SLI and SAD methods (both with a noise term), differences between the 3 sites can be seen. For the FINO3 data, the KNN method has better performance than the SLI and SAD methods for the MWS deviation, MWD deviation, and DE. For the FINO2 data, the KNN method performs best for the MWS deviation, but the SLI method with noise

term shows a lower DE and the SAD method shows a lower MWD deviation. For the Ijmuiden data, the KNN method shows the highest MWS deviation of all MCP methods. The distribution error is lower for the SLI method with noise term than for the KNN method. The KNN method shows the lowest MWD deviation.

## 4.2 Effect of gaps on long-term extrapolations

To analyze the impact of single gaps on long-term extrapolations, we follow the procedure outlined in Section 3.2.2: comparing

the long-term statistics of the gapped data with the long-term statistics of the original data. In Figure 5 we show the long-term mean wind speed, mean wind direction, and the Weibull parameters A and k depending on the gap starting date. The Weibull





**Table 4.** Performance statistics of each MCP method for the Ijmuiden site. Labels of statistics and MCP methods as described in Table 2.

| MCP method | MWS deviation [ms$^{-1}$] | WS DE [%] | MWD deviation [°] | WS RMSE [ms$^{-1}$] | WD RMSE [°] |
|---|---|---|---|---|---|
| SLI | **0.08** | 2.56 | - | **1.32** | - |
| SLI with noise term | **0.08** | **1.78** | - | 1.84 | - |
| KNN | 0.10 | 2.38 | **2.98** | **1.32** | **18.0** |
| SAD | - | - | 3.34 | - | 18.1 |
| SAD with noise term | - | - | 3.32 | - | 18.5 |

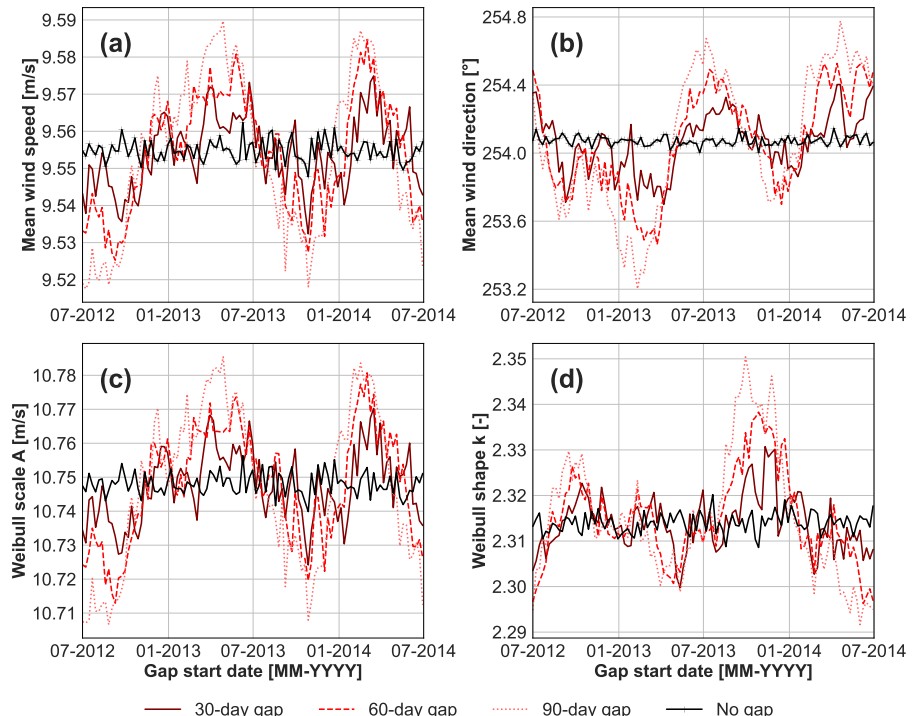

**Figure 5.** Long-term mean wind speed (subplot (a)), mean wind direction (subplot (b)), and Weibull parameters A (subplot (c)) and k (subplot (d)) of the original and gapped data depending on the gap starting date. Results for the original data in black, solid lines and for gap durations of 30, 60, and 90 days in maroon solid, red dashed, and light red dotted lines respectively. Long-term extrapolations are done with the Sector-wise Linear Interpolation (for wind speed) and Sector Average Deviation (for wind direction) MCP methods with noise term. Results for the FINO3 site.

parameters are obtained by fitting the wind speed distribution with a Weibull distribution function. Figure 5 shows results for the FINO3 site for a measurement time series with no gaps and for a time series with a gap of 30, 60, and 90 days.





For all long-term statistical parameters shown in Figure 5, the deviation between the gapped and the original parameters
varies depending on the gap starting date. These differences increase with increasing gap duration. The increase in the gap
effect is particularly pronounced for the starting dates for which the smallest (30-day) gaps already have their highest impact
on the LTE statistics. We obtain similar results for FINO2 and Ijmuiden data. Note that the variability of the LTE statistics of
the original data seen in Figure 5 is due to the noise term, as a new long-term extrapolation of the original data is done for each
gap starting date.

To generalize over all gap starting dates, the RMSE between gapped and original LTE statistics (see Section 3.2.2) is shown
in Figure 6 for all sites. The root mean squared distribution error between the original and the gapped LTE is shown instead of
the Weibull parameter deviations.

The left subplots in Figure 6 (subplots (a), (c), and (e)) show the results for a single gap with different gap lengths, introduced
as described in Section 3.3.1. An almost linear correlation between the statistics measuring the gap effect and the gap length
can be recognized. For all sites, a gap of 0 days affects the LTE due to the noise term. It leads to a mean wind speed RMSE of
roughly $0.005\,\mathrm{ms^{-1}}$, a mean wind direction RMSE of $0.05°$, and a distribution RMSE of 0.02 to 0.025%. For gaps longer than
0 days, there are differences between the sites which increase with increasing gap size:

- For the MWS (subplot (a)): results for the FINO2 and FINO3 sites show a higher mean wind speed RMSE between the
  LTE gapped and original data (around $0.037\,\mathrm{ms^{-1}}$ for 180-day gaps) than for the Ijmuiden site (roughly $0.02\,\mathrm{ms^{-1}}$ for
  180-day gaps).

- For the MWD (subplot (c)): results for the FINO3 and Ijmuiden sites show higher RMSE (approximately $0.7°$ for 180-
  day gaps) than for the FINO2 site (nearly $0.3°$ for 180-day gaps).

- For the wind speed distribution (subplot (e)): Few differences between the results for each site, Ijmuiden showing the
  lowest distribution RMSE (roughly 0.047% for 180-day gaps) and FINO3 showing the highest (nearly 0.065% for 180-
  day gaps).

These effects on the long-term extrapolation are negligible (<1%), considering that the 2-year mean wind speed in all
measurement sites is between 9.5 and $10\,\mathrm{ms^{-1}}$ and that the wind direction has a range of $360°$. This is true even for 180-day
gaps, which imply that roughly 25% of the measured data is missing.

Subplots (b), (d), and (f) on Figure 6 show the RMSE of the long-term statistics over all introduced 30-day gap combinations
depending on the combined gap length (for more information on multiple gap introduction, see Section 3.3.2). As well as for the
single gap case, a linear relationship between the gap effect and combined gap length can be seen. Furthermore, the differences
between the sites are similar for the multiple and the single gap case. Nevertheless, when measured using the RMSE between
the gapped and the original long-term statistics, the effect of multiple gaps is lower than the effect of the single gap for all sites
and gap lengths.

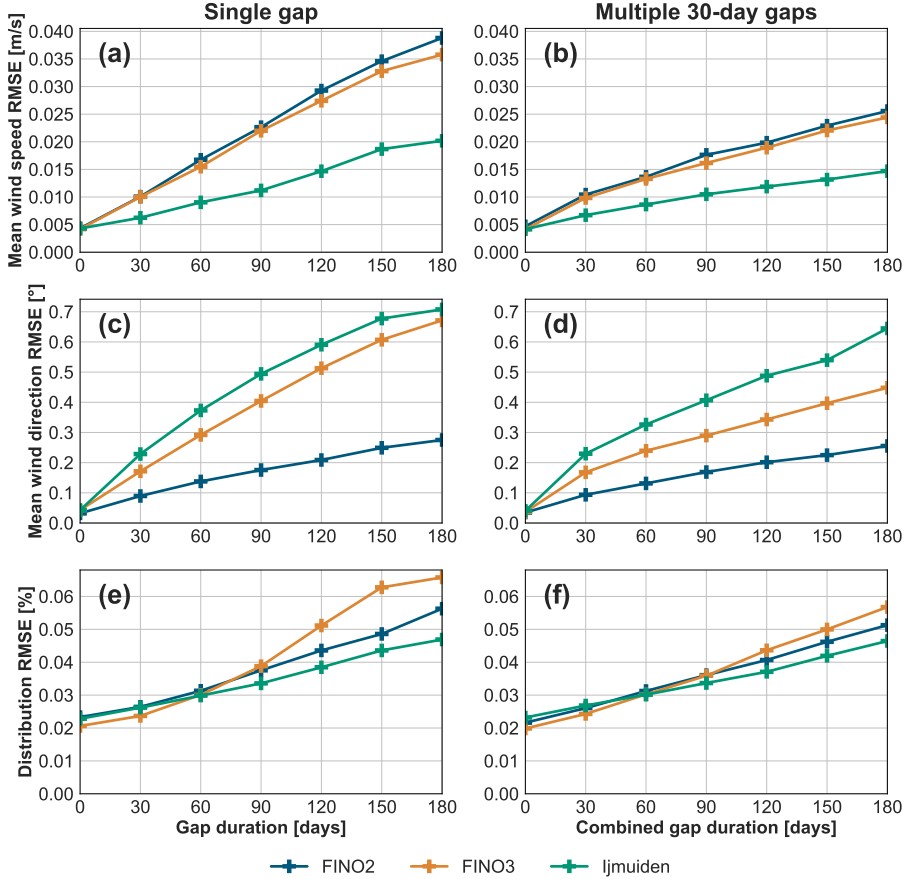

**Figure 6.** Long-term mean wind speed (subplots (a) and (b)), mean wind direction (subplots (c) and (d)), and distribution (subplots (e) and (f)) RMSE between the original and gapped data calculated over all introduced gaps for each gap duration. Results for one gap with shifting start date on the left (subplots (a), (c), and (e)) and for multiple gaps with random starting dates on the right (subplots (b), (d), and (f)). Long-term extrapolations are done with the Sector-wise Linear Interpolation (for wind speed) and Sector Average Deviation (for wind direction) MCP methods with noise term. Results for the FINO2 site in blue, the FINO3 site in orange, and the Ijmuiden site in green color.

## 4.3 Impact of gap filling on long-term extrapolations

345

In a final step, we use single gaps (introduced as described in Section 3.3.1) to evaluate the effect of gap filling on long-term extrapolations with the method described in Section 3.2.2. Figure 7 summarizes the results of this investigation by showing the variation of the RMSE calculated over all gap starting dates between LTE statistics of the original and the gapped data (continuous lines) and between the original and the filled data (dotted lines).



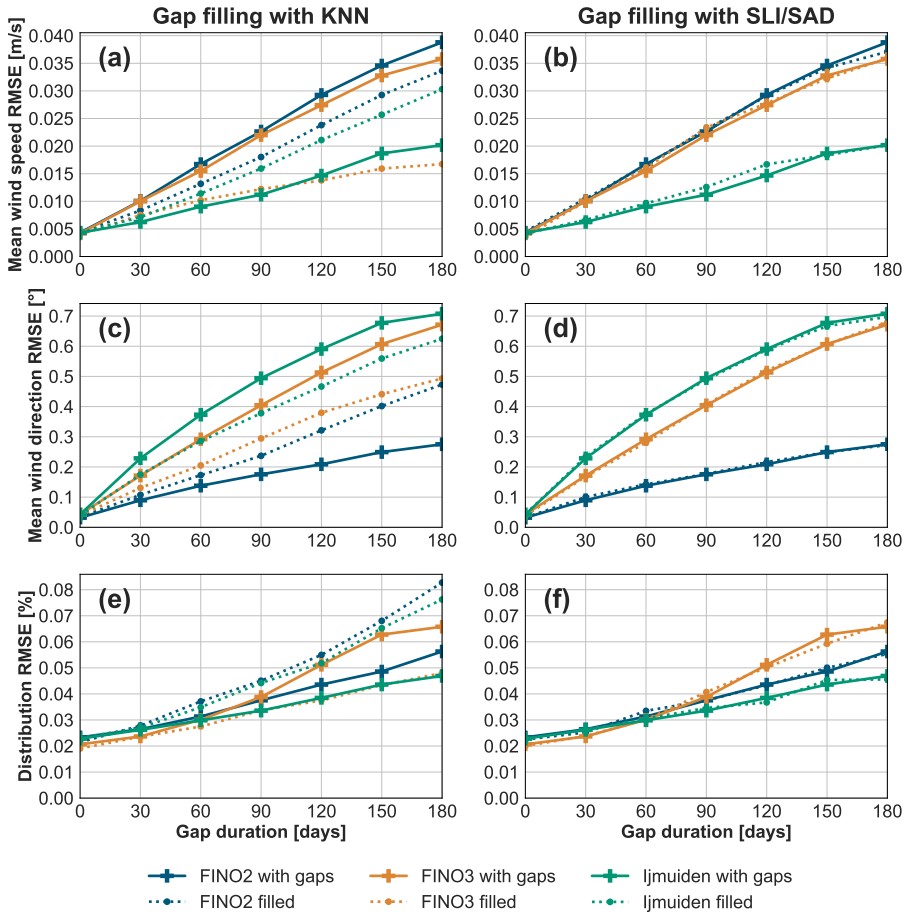

**Figure 7.** Long-term mean wind speed (subplots (a) and (b)), mean wind direction (subplots (c) and (d)), and distribution (subplots (e) and (f)) RMSE between the original and gapped data (continuous lines) and between the original and filled data (dotted lines). RMSE calculated over all introduced gaps for each gap duration for single gaps with shifting start date. Gap filling is done with the KNN MCP method on the left (subplots (a), (c), and (e)) and with the Sector-wise Linear Interpolation (SLI, for wind speed) and Sector Average Deviation (SAD, for wind direction) MCP methods with noise term on the right (subplots (b), (d), and (f)). Long-term extrapolations are done with the SLI and SAV with noise term. Results for the FINO2 site in blue, the FINO3 site in orange, and the Ijmuiden site in green color.





The subplots on the right in Figure 7 (subplots (b), (d), and (f)) show the effect of filling the wind speed gaps with the SLI MCP method and the wind direction gaps with the SAD MCP method. It can be seen that filling the gaps with these methods does not affect the long-term extrapolations for any of the sites and analyzed parameters. The noise term used when filling and extrapolating causes the minimal deviations between the gapped and the filled LTE.

The left subplots in Figure 7 (subplots (a), (c), and (e)) show an approximately linear correlation between the gap effect and the gap length for the KNN-filled gaps for all sites. According to these subplots, filling with the KNN method has a different impact on the long-term extrapolations depending on the site and the evaluated statistic:

- For the FINO3 data: filling with the KNN method reduces the RMSE for all metrics.

- For the FINO2 data: filling with the KNN method reduces the mean wind speed RMSE, but increases it for the mean wind direction and the wind speed distribution.

- For the Ijmuiden data: filling with the KNN method reduces the mean wind direction RMSE, but increases it for the mean wind speed and wind speed distribution.

It must be noted that the sites and statistics for which filling with the KNN method reduces the gap effect on the long-term extrapolations are the same as the cases found in Section 4.1 for which the KNN method shows a better performance than the SLI or SAD methods with noise term. We discuss this and other considerations on the results in Section 5.

## 5 Discussion

When comparing the performance of the MCP methods in Section 4.1, we find that they perform differently depending on the site and the metric used. The KNN MCP method excels when estimating each single data point because the selection of K is optimized for that purpose (reduction of the RMSE between measurement and prediction, see Section 3.1.3). This agrees with the results shown by Schwegmann et al. (2023). The methods with noise term perform worst by that measure, as they introduce an artificial error on each data point. Nevertheless, the addition of the noise term does not affect the mean wind speed and direction values, as the artificial error is averaged to 0.

Regarding the wind speed distribution, Hanslian (2017) points out that type 1 MCP methods such as linear regression are most suited for predicting wind speed time series, but distort distributions. Tables 2 to 4 show a lower distribution error to the original data when the noise term is added than when it is not. Therefore, we conclude that the noise term partly compensates for the distortion of the wind speed distribution induced by the linear regression for the cases tested in the present work. The noise term consists of random samples of a Gaussian distribution added to each wind speed value. Therefore, the distribution distortion is only compensated if the distribution of the measured data used for testing is more similar to a Gaussian distribution than the distribution of the predicted data. This is the case for the investigated sites, but the contrary is possible for other sites or periods, for which the addition of the noise term would increase the distribution distortion. Therefore, we dismiss the Gaussian noise term as a universal solution for this issue and introduce the KNN method as an alternative. This is an analog MCP method



by the criteria of Hanslian (2017). These methods combine type 1 and type 2 features and can also distort distributions, as the found analogs (neighbors in the case of KNN) tend to have values closer to the mean than further away from the mean (Hanslian, 2017). No clear choice between the KNN method and the linear interpolation with noise term can be derived from the results in Section 4.1 for predicting wind speed distributions. Given the error potentially introduced by the noise term, we recommend

using either distribution-based MCP methods such as matrix methods (see Hanslian (2017)) or analog methods such as KNN when predicting wind speed distributions. All methods studied in Section 4.1 have similar performances for predicting wind speed and direction averages. We assume that the linear interpolation and sector average deviation MCP methods are sufficient for this purpose, although more complex methods might give slightly better results. If an accurate point-by-point prediction is the aim, the RMSE between the predicted and the testing measured data is to be reduced. Given the results obtained in Section

4.1 and in the work of Schwegmann et al. (2023), we consider that training a KNN or other machine learning models such as shown by Schwegmann et al. (2023) is the best solution.

It must be noted, that the statistics listed in Tables 2 to 4 are an averaged value over the results obtained for each introduced gap. Therefore, the comparison between the performance statistics can be different for one specific gap than for the averages shown. The classification of MCP methods based on their performances is only valid for the data, gap introduction procedure,

and metrics used in the present work.

In the present study, we apply the metric proposed by Gottschall and Dörenkämper (2021) to describe the effect of gaps on long-term extrapolations with the linear regression MCP method with noise term. The different results for the long-term extrapolations are due to using different wind sector divisions, regression functions, and target periods than Gottschall and Dörenkämper (2021). Nevertheless, the finding of the low and the seasonally changing gap effect aligns between Gottschall

and Dörenkämper (2021) and the present study. We build on the study of Gottschall and Dörenkämper (2021) by increasing the gap length and by introducing multiple gaps. The increase of the gap effect with increasing gap size is expected since the gapped and the original data sets increasingly differ. Nevertheless, even 180-day gaps (corresponding to approximately 75% availability) show a low effect. These gaps only cause a $0.037\,\mathrm{ms}^{-1}$ long-term mean wind speed error on average for FINO2 and FINO3. $0.037\,\mathrm{ms}^{-1}$ are roughly 0.38% of the mean wind speed measured. This added uncertainty is minor compared to

other long-term uncertainty sources such as the uncertainty of the wind speed measurement, which can surpass 3% (Pulo et al., 2021). We show that the gap effect is even lower for multiple gaps with the same combined gap length. We assume that this is because the single gap is longer and more likely to cut out an entire season so that climatic effects specific to that season are ignored.

Given the low effect of gaps on long-term extrapolations, we recommend lowering the requirements on data availability of

over 80 or 90% given in current guidelines (MEASNET, 2016; FGW, 2020) for offshore measurement campaigns. This will reduce the cost of obtaining the on-site wind data while not impacting the wind resource assessment significantly. Furthermore, we align with the method to assess gap effects on long-term extrapolations recommended by Gottschall and Dörenkämper (2021) and used in the present investigation. This can be a valid method not only for further investigations on the effect of gaps but also for estimating the effect of a real gap on a wind resource assessment scenario.



Even though gaps have a low effect on the long-term extrapolations in the investigated cases, we show how gap filling can change the gap effect on the LTE. When filling with the linear regression method, no difference between the filled and gapped long-term extrapolations can be seen. This is because the same MCP method and reference data sets are used for filling and extrapolating. In this case, the training measurement data and the correction function of the MCP method are the same for filling the gaps and for extrapolating the gapped data set. Therefore, the correction function of the long-term extrapolation

already contains the information that is obtained by filling the gaps and stays unchanged when the gap-filling data is factored in.

Gap filling might decrease the bias of the LTE caused by gaps when done with an MCP method other than the extrapolating method. We investigate this using the KNN method for filling and the linear interpolation (for wind speed) and sector average deviation (for wind direction) MCP methods with noise term for extrapolating. With this setting, gap filling does not always

mitigate the gap effect on the LTE (see subplots A, C, and E in Figure 7). Nevertheless, the gap effect is mitigated for the predicted parameters and sites, for which KNN has a better performance for gap filling than the linear interpolation and sector average deviation MCP methods with noise term (see Section 4.1). We presume that the gap-filling performance and the reduction of the gap effect on the LTE correlate for each predicted statistical parameter and site. This can be investigated further with various gap-filling and long-term extrapolating MCP methods and different data sets. However, neither the performance

of a gap-filling method nor the reduction of the gap effect on the LTE can be calculated for a real gap in the measured data. Therefore, introducing and analyzing the effect of filling artificial gaps is the only way to estimate the effect of filling real gaps. Hereby, the period cut out by the real gaps has to be considered (periods with very high or low wind speed might have the largest effects when cut out). Artificially cutting out periods with similar wind climate as the period cut out by the real gaps might give an insight into how the real gap affects the long-term extrapolation. If redundant measurements are available (for

example, data from another height or a nearby deployed floating lidar system), gaps should be filled with those redundancies as reference instead of reanalysis data. As less correction is needed for reference data from redundant measurements, we expect gap-filling to reduce the uncertainty of long-term extrapolations in these cases.

## 6    Conclusions

Since data gaps are common in offshore wind measurements, multiple guidelines limit the proportion of missing data allowed.

Therefore, wind-measuring stakeholders resort to expensive wind measurement campaigns due to redundant, monitored systems and prolonged measurement periods. One of the goals of the present study was to find out whether the current industry requirements on measurement data availability are justifiable or too conservative for offshore measurement campaigns. To answer this question, we built on the research of Gottschall and Dörenkämper (2021) and investigate the effect of gaps on long-term (20-year) extrapolations in multiple settings. We analyzed the effects of gaps with various lengths and amounts

for the same sites as investigated by Gottschall and Dörenkämper (2021): met mast measurements from FINO2, FINO3, and Ijmuiden from the period between 2012 and 2014. Throughout the present study, we used the linear regression MCP method with ERA5 as reference data for long-term extrapolating. The metric we used for evaluating the gap effect on the extrapolations





is the RMSE between the measured and the gapped long-term mean wind speed, mean wind direction, and distribution over all introduced gaps. The study on the gap effect on long-term extrapolations yielded the following results:

– In alignment with the results of Gottschall and Dörenkämper (2021), we found that gaps have a minor impact on long-term extrapolations. Even for the availability of 75%, the RMSE between the gapped and the original long-term mean wind speed does not surpass $0.04\,\mathrm{ms^{-1}}$ for any of the analyzed sites. We obtained similar results for the long-term mean wind direction and wind speed distribution.

– A single gap has more effect on the long-term extrapolation than multiple gaps with the same combined length. We assume this is because a single wind data gap is more likely to cut out a climatic event than several shorter gaps spread out through the time series.

In addition to investigating the gap effect on long-term extrapolations, we analyzed the possibility presented by Gottschall and Dörenkämper (2021) of filling the gaps to decrease their effect on the extrapolation. We introduced the KNN, linear interpolation, and sector average deviation MCP methods for filling data gaps and compared their performance. Furthermore,
we evaluated the relationship between the performance of a gap-filling MCP method and its reduction of the gap effect on a long-term extrapolation. We obtained the following results:

    – The linear interpolation MCP method distorts wind speed distributions. Adding a Gaussian noise term reduces the distribution distortion for sites with Bell-shaped distributions. However, this reduces the accuracy of predicting a wind speed time series value-by-value.

– The KNN MCP method shows good results (compared to the linear regression and sector average deviation methods) when predicting mean wind speed, mean wind direction, wind speed distributions, and wind speed and wind direction time series.

    – Filling the gaps does not impact the long-term extrapolation for filling and the extrapolating processes done with the same MCP method and reference data.

– The gap effect on long-term extrapolations is reduced through filling with the KNN method for the same parameters and locations for which the KNN method outperforms the linear interpolation and sector average deviation MCP methods in terms of gap-filling.

According to the present findings, the minimum data availability acceptable for an offshore measurement campaign should be lower than the 90% currently demanded by most standards. As gaps have a low effect on extrapolations, gap-filling does not
significantly reduce the gap effect. Nevertheless, data from a different measurement height or a nearby deployed device is often available. The nearby taken data has a lower uncertainty and better correlation with the analyzed measurement than modeled data such as ERA5. Therefore, we expect that using the nearby data as reference data for the gap-filling MCP method reduces the effects of gaps on the long-term extrapolation, even if this effect is mild.



*Data availability.* The ERA5 reanalysis data was downloaded from the Copernicus Climate Change Service Climate Data Store (https://cds.-
climate.copernicus.eu/cdsapp#!/dataset/reanalysis-era5-single-levels?tab=overview). The met mast data can be accessed for scientific pur-
poses by contacting BSH (for FINO2 and FINO3) and TNO (for Ijmuiden).

*Author contributions.* MJA analyzed and processed the data and implemented the methods. WW contributed to several parts of the used
code and participated in the data and result interpretation. JG provided the wind data and supervised the study. MJA wrote the first draft of
the manuscript and all authors were engaged in reviewing it. All authors approve the content and agree to be held accountable for it.

*Competing interests.* At least one of the (co-)authors is a member of the editorial board of Wind Energy Science.

*Acknowledgements.* This research was partly carried out in the framework of the project Digitale Windboje (ref. no. 03EE3024) funded
by the German Federal Ministry for Economic Affairs and Climate Action (BMWK). We appreciate the technical and academic advice of
Sandra Schwegmann and Martin Dörenkämper. Furthermore, we thank BSH for providing access to the data measured at FINO2 and FINO3,
TNO for the data of the IJmuiden met mast, and ECMWF for providing open access to the ERA5 data.



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
