# Peer review of "Understanding the impact of data gaps on long-term offshore wind resource estimates"

_Wind Energy Science, 2023_

## Author Comment (AC2)

**"Understanding the impact of data gaps on long-term offshore wind resource estimates"**
*Rev v1*
Martin Jonietz Alvarez, Warren Watson, and Julia Gottschall
Wind En. Sci. https://doi.org/10.5194/wes-2023-127, 2023

**Authors response to reviewer comments**

We thank the referees for their time and effort dedicated in reviewing our manuscript. We highly value their feedback and critiques, and we have carefully considered their comments to enhance and clarify our work.

Presented below are our responses to each of the referees' remarks, addressed point by point. Each comment from the referee is presented in bold font, followed by our corresponding response. Where applicable, we have included the revised excerpt from the manuscript (highlighted in blue).

**Anonymous Referee, Referee #1**

**Referee #1 comments and questions**

1) **The metrics used to evaluate the MCP method performance for wind redirection data may bring some deviation in the Northerly direction sector, such as 359°and 1°. The authors are advised to address this issue more clearly.**

   Clarifications added in:

   - Chapter 3.1.4: "...The wind direction differences are in the range of -180° to 180°, as they are calculated using the shortest path in the 360° circle..."
   - Chapter 3.2.2: "...The wind direction differences are calculated using the shortest path in the 360° circle..."

2) **It is recommended that the different curves in Figure 5 be distinguished by colors instead of line types.**

   We changed the figure as you suggested. We chose a colorblind-differentiable color palette that is different to the palette in Figures 6 and 7. This is to avoid confusion between the long-term values for each gap shown in Figure 5 and the RMSE of the deviations between the long-term values shown in Figures 6 and 7. Here is the result:

[Figure]

Figure 1: Long-term mean wind speed (subplot (a)), mean wind direction (subplot (b)), and Weibull parameters A (subplot (c)) and k (subplot (d)) of the original and gapped data depending on the gap starting date. Results for the original data in black, solid lines and for gap durations of 30, 60, and 90 days in purple, teal, and light green lines respectively. Long-term extrapolations are done with the Sector-wise Linear Interpolation (for wind speed) and Sector Average Deviation (for wind direction) MCP methods with noise term. Results for the FINO3 site.

**Anonymous Referee, Referee #2**

**Referee #2 specific comments and questions**

1) **P1L10-11: You say it's lower than expected, but you don't say what you expected and why you expect that. You should help the reader set what reasonable expectations are.**

When the availability figures of the standards are exceeded, we expect gap-induced deviations of the long-term mean or the distributions larger than the measurement uncertainty or other uncertainty components. We clarify that in the abstract with the following updated text:
"... Current standards demand high data availability (80 or 90%) for wind measurement campaigns, so we expect that the effect of missing data on the uncertainty of long-term extrapolations is on the same order of magnitude as other uncertainty components such as the measurement uncertainty

or the inter-annual variability. Nevertheless, our results show that gap effects are considerably lower than the other uncertainty components….”

2) **P1L11-12: Throughout the paper, you focus mostly on the mean outcomes, but is that the most important measure to consider? what happens to the results if you consider the uncertainty? the P75, P90 results? I would expect many readers to want to know about the worst outcomes.**

We now include the analysis of P95. We included the description of the metric in Section 3.2.2: "… To generalize the results over all gaps with the same gap duration introduced, we use two metrics:

- The RMSE between the gapped and the original long-term extrapolated MWS, MWD, and distributions. We adopt this metric from Gottschall and Dörenkämper (2021) to assess the average gap effect on long-term extrapolations.

- The P95 of the absolute deviation between the gapped and the original long-term extrapolated MWS, MWD, and distributions. We use this metric to highlight the 5% highest of all analyzed gap effects.

We use the same metrics for the deviations between the filled and original long-term statistics to analyze the effect of gap filling on long-term extrapolations….”

The P95 gap effects on long-term extrapolations are roughly double the results of the RMSE analysis. We now present and discuss the results for P95 along with the results for the RMSE. The main conclusions of the study remain unchanged.

3) **P1L12-13: Why would I use a slightly worse method to do long-term correction, than the one I use for gap-filling?**

In theory, there is no reason to do this. Nevertheless, we assume that industry stakeholders will be reluctant to use the KNN method for long-term extrapolations, as it is not validated in this context. We removed the remark from the abstract as the effect of gaps and gap filling is very slight anyway. We still elaborate on this case in the discussion section.

4) **P2L38-39: How many months is "various months"?**

We now include an example to be more specific on that statement: "…An overview of offshore met mast measurements by Meyer and Gottschall, 2022 shows that gaps of 2 or 3 months are a common phenomenon, and that longer gaps are possible as well…”

5) **P3L78-79: I would rephrase the "exact" part here. It sounds as if mast measurements have no error or uncertainty.**

We removed the remark that was stating that met mast measurements are "exact".

6) **P5L113-114: Did you resample to 1H by averaging, or simply taking the 10 Min values every hour?**

We take the 10 Min value at every hour. We adapted the text so that it is more clear: "…As the reanalysis data used as reference for the MCP methods is only available with hourly resolution, we only use the 10 minute met mast measurement values that are stamped at whole hours…."

7) **P7L186: How is the circular nature of wind direction treated in the KNN model? e.g. is a wind direction of 359.9° a close neighbor of a wind direction of 0.1°?**

Explanation added: "…Cyclical parameters such as the wind direction are split into their sine and cosine values to include them in the feature space…"

8) **P10-11 Gap generation: As you allude, the worst-case gaps are probably large contiguous gaps near the extremes of the annual cycle, i.e. summer and winter in your cases. Did any of the gap-generation methods result in large gaps in summer or winter in both years? having measurements from one summer can probably alleviate the problems from large gaps in the other year, but if it's missing from both it would be a problem. Perhaps this is out of the scope of your study? It would be good if you mentioned that in the paper in that case. Even so, it would be interesting to know what the results are for the case where both summers (JJA) are removed or both winters (DJF), which would probably represent worst-case scenarios for a 180-day gap. In general, I am missing some more comments/discussions about the annual cycle and the effect of generating gaps in the extremes of the cycle. Total availability (e.g. 75\%) is not as interesting as the distribution of availability.**

We agree that the distribution of the gaps may have a significant impact on the long-term extrapolation. The effect of gap seasonality on long-term extrapolations would be therefore an interesting follow-up study but is outside of the scope of our work. We include now a paragraph in the discussion section (Section 5) addressing this topic: "…Figure 5 shows an example of the seasonality of the gap effect for FINO3. In this case, the long-term extrapolations are most sensitive to gaps covering the spring and autumn months. We could observe a similar seasonality for the other analyzed sites. The results of Gottschall and Dörenkämper also show the largest impact of gaps when they cover spring and autumn months for all sites. Nevertheless, the differences between the seasons in that study are slight and therefore non

conclusive because only 30-day gaps are considered. As the sensitivity of long-term extrapolations to the season of the gap is not the object of the present study, we do not investigate further on this topic. However, a follow-up study with a more extensive analysis might be of interest to the stakeholders involved in wind resource assessment…."

**Referee #2 technical corrections**

1) **P6L148: Missing number. What Type?**

Corrected (Type1)

**Further changes**

1) **Update of source *Rouholahnejad, F., Santos, P., Hung, L.-Y., and Gottschall, J.: Machine learning for predicting offshore vertical wind profiles, Journal of physics, submitted and accepted for publication, 2023.***

Now the final accepted manuscript is cited, including a DOI